# Autocrine TGF-β in Cancer: Review of the Literature and Caveats in Experimental Analysis

**DOI:** 10.3390/ijms22020977

**Published:** 2021-01-19

**Authors:** Hendrik Ungefroren

**Affiliations:** 1First Department of Medicine, University Hospital Schleswig-Holstein, Campus Lübeck, D-23538 Lübeck, Germany; Hendrik.ungefroren@uksh.de; 2Clinic for General Surgery, Visceral, Thoracic, Transplantation and Pediatric Surgery, University Hospital Schleswig-Holstein, Campus Kiel, D-24105 Kiel, Germany

**Keywords:** autocrine signaling, cancer, feedback loop, transforming growth factor-β

## Abstract

Autocrine signaling is defined as the production and secretion of an extracellular mediator by a cell followed by the binding of that mediator to receptors on the same cell to initiate signaling. Autocrine stimulation often operates in autocrine loops, a type of interaction, in which a cell produces a mediator, for which it has receptors, that upon activation promotes expression of the same mediator, allowing the cell to repeatedly autostimulate itself (positive feedback) or balance its expression via regulation of a second factor that provides negative feedback. Autocrine signaling loops with positive or negative feedback are an important feature in cancer, where they enable context-dependent cell signaling in the regulation of growth, survival, and cell motility. A growth factor that is intimately involved in tumor development and progression and often produced by the cancer cells in an autocrine manner is transforming growth factor-β (TGF-β). This review surveys the many observations of autocrine TGF-β signaling in tumor biology, including data from cell culture and animal models as well as from patients. We also provide the reader with a critical discussion on the various experimental approaches employed to identify and prove the involvement of autocrine TGF-β in a given cellular response.

## 1. Introduction

The transforming growth factor-βs (TGF-βs), TGF-β1, 2 and 3, are secreted polypeptides that signal via two types of membrane serine/threonine kinase receptors, type II (TβRII) and type I (TβRI), and intracellular Smad effectors [1,2,3]. TGF-β1, the most common isoform in human cancers [3], inhibits proliferation and induces apoptosis in various normal and premalignant human epithelial cells and its essential signaling intermediates, i.e., TβRII and Smad4, are therefore considered tumor suppressors. The anti-oncogenic function of this pathway is supported by the frequent occurrence in cancer cells of genetic and epigenetic alterations that abolish its growth-inhibitory function. In addition, various oncogenes directly hijack the TGF-β/Smad pathway to favor tumor growth. On the other hand, all advanced human tumors overproduce TGF-β, whose autocrine and paracrine actions in most instances promote tumor growth, invasion, and metastasis [3,4]. TGF-β is a powerful inducer of epithelial-mesenchymal transition (EMT), a differentiation switch that is required for transitory invasiveness of carcinoma cells, the generation of cancer stem cells (CSCs), and phenotypic plasticity, eventually resulting in tumor heterogeneity and resistance to standard chemotherapies [5,6]. Tumor-derived TGF-β acting on stromal fibroblasts generates cancer-associated fibroblasts (CAFs), remodels the tumor stroma and eventually induces expression of mitogenic and survival signals towards the carcinoma cells, while TGF-β acting on endothelial cells and pericytes regulates (neo)angiogenesis [3]. TGF-β also suppresses proliferation and differentiation of lymphocytes, including cytolytic T cells, natural killer cells and macrophages, thus preventing effective eradication of the developing tumor by the host immune system [7]. Hence, TGF-β signaling is intimately involved in nearly all aspects of tumor development [8].

Autocrine signaling is defined as the production and secretion of an extracellular mediator, i.e., a growth factor or cytokine, by a cell followed by the binding of that mediator to receptors on the surface of the secreting cell to initiate signaling. In a less strict sense this definition will also include interactions with receptors on neighboring cells when these cells are of the same type with respect to differentiation or function such as in an epithelial lining (Figure 1). In contrast, paracrine signaling occurs between different types of cells, i.e., an epithelial/carcinoma cell and a CAF or a T cell (Figure 1). For several growth factors there is now evidence that the (mitogenic) signal may be transduced without factor secretion. In these instances, the growth factor appears to interact with its receptor within the cell, i.e., in cultured hepatocytes, creating a “private” autocrine or intracrine loop [9,10], in contrast to the classical “public” autocrine loop [9] (Figure 1).

Autocrine stimulation often operates in autocrine loops, a type of interaction, in which a cell produces a mediator, for which it has receptors that upon activation promote—in a direct or indirect fashion—expression of the same mediator, allowing the cell to repeatedly autostimulate itself. As an alternative to this positive feedback or feedforward loop, the mediator may balance its own expression by inducing, or suppressing, a second factor that regulates this same mediator in an opposite fashion to provide negative feedback.

Whereas loss or attenuation of TGF-β signaling is permissive for transformation, blocking receptor function in metastatic breast cancer (BC) cells has been shown to inhibit survival, EMT, invasiveness, and metastatic dissemination, suggesting that TGF-β promotes tumor development and malignancy through autocrine and/or paracrine mechanisms [11]. In contrast, autocrine TGF-β (aTGF-β) may also attenuate tumor progression, i.e., by its ability to prevent escape from oncogene-induced senescence (OIS). Based on this observation, aTGF-β has been suggested to be part of a cellular anti-transformation network [12], adding further support to its dual nature in cancer biology.

Although derived from the same genes, exogenous and endogenously produced TGF-β may have different and sometimes antagonistic effects depending on cell type, context, tumor stage and effective concentration or duration of exposure. The outcome is further complicated by the complex process of TGF-β activation [3] and the unique mode of receptor activation, which allows the cells to fine-tune their sensitivity to exogenous ligand in a spatial, temporal and concentration-dependent manner [13]. In addition, TGF-β1 can induce its own expression [14,15] and that of its receptors [16], while the receptors, in addition to canonical Smad signaling, can activate several pathways commonly associated with tyrosine kinase receptors [17], with competition among them for access to receptors [18]. Finally, these signaling pathways exhibit extensive crosstalk with tyrosine kinase signaling, i.e., from Ras (see Section 3.8). Moreover, cell lines derived from human microsatellite unstable colorectal cancers (MSI-H CRCs) with truncating mutations in *TGFBR2*, were not completely refractory to TGF-β, despite the lack of functional TβRII. Since these cells remained sensitive to signaling by endogenously produced TGF-β, at least with respect to invasive abilities [19] (see Section 2.1), aTGF-β and exogenous TGF-β may have different receptor requirements.

In this work, we have reviewed various observations on the involvement of aTGF-β in tumor biology, including data from cell culture and animal models as well as from patients, and various cancer types. We close this review with some general considerations on autocrine signaling and a critical discussion on the various experimental approaches employed to identify and prove the involvement of aTGF-β in a given cellular response. We believe that in some studies this issue has not been adequately addressed and definite conclusions as to the participation of aTGF-β must, therefore, remain preliminary.

## 2. Autocrine TGF-β in Cancer-Associated Processes

### 2.1. Cell Proliferation and Apoptosis

A mentioned above, some cell lines derived from MSI-H CRCs were not completely refractory to TGF-β, despite the lack of functional TβRII [19]. Re-expression of TβRII in HCT116 cells restored aTGF-β signaling and reduced proliferation due to induction of p21^CIP1^ but failed to restore growth-inhibitory responses to exogenous TGF-β, indicating that aTGF-β regulates the cell cycle through a pathway different from exogenous TGF-β [4]. Additionally in MSI-H CRC cell lines, Baker et al. provided evidence for an independent function of TβRI in signaling by aTGF-β. While these cells were capable of binding exogenous TGF-β, a fraction of them failed to respond with signaling activity. The use of a specific inhibitor of TβRI, however, revealed that these remained sensitive to signaling by endogenously produced aTGF-β as evidenced by constitutive activation of Smad2 and repression of extracellular signal-regulated kinase (ERK) signaling. Autocrine signaling via TβRI also promoted the invasion of MSI-H CRC cells to a similar extent as that seen in their non-MSI-H counterparts but failed to impact proliferation [19], supporting the idea that endogenous/aTGF-β and exogenous/paracrine TGF-β can mediate different cellular functions.

Treatment of human triple-negative breast cancer (TNBC) MDA-MB-231 cells with TGF-β neutralizing antibodies, recombinant human soluble (s) TβRIII, or ectopic expression of sTβRIII, inhibited both anchorage-dependent and -independent cell growth and induced apoptosis in vitro and in vivo, suggesting specific antagonization of aTGF-β signaling and its requirement for the growth and survival of MDA-MB-231 cells [20]. A follow-up study using abrogation of aTGF-β signaling by expression of a dominant-negative (dn) mutant of TβRII or the treatment with a small-molecule TβRI inhibitor significantly increased apoptosis in MCF-7 cells but not in untransformed human mammary epithelial cells (HMECs), suggesting that in transformed BC cells aTGF-β signaling can enhance cell survival by maintaining high and low levels, respectively, of active ERK and p38 [21]. Along the same lines, systemic administration of a sTβRII:Fc fusion protein to mouse mammary tumor virus-polyomavirus middle T antigen (MMTV-PyVmT) transgenic mice increased apoptosis in primary tumors and reduced tumor cell motility, intravasation, and lung metastases [22]. Hoshino and colleagues confirmed that aTGF-β signaling in certain highly metastatic BCs promotes cell survival via inhibition of apoptosis. In addition, they demonstrated by inhibiting endogenous TGF-β signaling with a TβRI kinase inhibitor in cells cultured serum-free that aTGF-β suppressed the expression of the pro-apoptotic protein, Bim [23]. Interestingly, inhibition of aTGF-β signaling in MDA-MB-231 cells reduced p21^CIP1^ expression and cell growth, suggesting that aTGF-β signaling is required to sustain p21^CIP1^ levels for positive regulation of cell growth [24].

Human hepatocellular carcinoma (HCC) cells stably expressing a Smad2 mutant, in which the serine residues of the C-terminal SSXS motif were changed to alanine, demonstrated impaired Smad2 signaling and were resistant to growth inhibition by (exogenous) TGF-β. Interestingly, however, forced expression of this mutant induced aTGF-β secretion, which enhanced signaling through Smad3 and Smad4, and up-regulated plasminogen activator inhibitor-1 (PAI-1) and vascular endothelial growth factor (VEGF). This is consistent with TGF-β regulating its own synthesis and provides an example for functional specification of signaling by Smad2 and Smad3, which appear to antagonistically regulate aTGF-β production in human HCC [25].

### 2.2. EMT, Stemness and Cell Motility

There is increasing evidence that after cells have lost their sensitivity to TGF-β-mediated growth inhibition, aTGF-β signaling promotes tumor cell motility and invasiveness. In MDA-MB-231 cells stably expressing dnTβRII basal migration was impaired but could be restored by reconstitution of TGF-β signaling with a constitutively-active (ca) TβRI but not by reconstituting Smad signaling. The caTβRI^T204D^ mutant does not rely on exogenous ligand in order to be able to signal. From the observation that introducing this mutant into cells expressing dnTβRII restored their migratory response and was associated with an increase in AKT and ERK but not Smad2 phosphorylation, the authors concluded that aTGF-β signaling can promote cell motility in a Smad-independent manner [26].

The *Neu* (erbB2) proto-oncogene product is the murine ortholog of HER2 and like epidermal growth factor receptor (EGFR; erbB1) belongs to the erbB family of receptor tyrosine kinases. To determine if the Neu is dominant over TGF-β, Dumont et al. crossed MMTV-Neu mice with MMTV-TGF-β1(S223/225) mice expressing active TGF-β1 in the mammary gland. The bigenic tumors and their metastases were less proliferative than those occurring in MMTV-Neu mice, however, bigenic tumors exhibited lower apoptotic scores and were more locally invasive. Mice harboring bigenic tumors contained a greater number of circulating tumor cells and lung metastases as well as higher levels of activated Smad2, Akt, Erk, p38, Rac1 and more vimentin in the tumor tissues in situ than tumors expressing Neu alone. These changes were inhibited by sTβRII:Fc, suggesting they were activated by aTGF-β. The data indicate that Neu does not abrogate aTGF-β1-mediated antiproliferation but can synergize with aTGF-β1 in accelerating metastatic tumor progression [27]. In a follow-up study, Muraoka-Cook and colleagues addressed the role of TGF-β in the progression of established tumors. To spare the inhibitory effects of TGF-β on early transformation, they generated triple transgenic mice with doxycycline-inducible regulation of active TGF-β1 expression in mammary tumor cells transformed by the PyMT. Conditional induction of TGF-β1 for 2 weeks strongly increased lung metastases without detectable effects on primary tumor cell proliferation or tumor size. Doxycycline-induced active TGF-β1 protein and nuclear Smad2 were restricted to cancer cells, suggesting a causal association between aTGF-β and increased metastasis. The selective effect of aTGF-β on invasion and metastasis was subsequently confirmed by the reverse approach, antisense-mediated inhibition of TGF-β1 in the tumor cells, indicating that the induction and/or activation of TGF-β in hosts with already established TGF-β-responsive cancers can rapidly accelerate metastatic progression [28].

Another study investigated the role of aTGF-β signaling in the survival and metastatic potential of mammary CSCs, utilizing a novel murine mammary cell line, NMuMG-ST, which acquired CSC phenotypes during spontaneous transformation of the parental cell line, NMuMG. In NMuMG-ST cells, aTGF-β signaling promoted anchorage-independent growth, resistance to serum deprivation-induced apoptosis, EMT, sphere formation, and the expression of stem cell markers. Upon injection into mice, these cells underwent apoptosis and generated less lung metastases than control cells, while the sizes of xenograft tumors were not different, indicating that aTGF-β signaling is involved in the maintenance and survival of murine BC stem cells and their enhanced metastatic ability [29].

During tumor pathogenesis, changes in cell phenotypes are induced by contextual signals that epithelial cells receive from the tumor microenvironment (TME) [6] and that include TGF-β and Wnt ligands. Their signaling pathways collaborate in several feedback loops to activate the EMT program and thereafter function in an autocrine fashion to maintain the resulting mesenchymal state [30]. Tian et al. have constructed a mathematical model for the EMT core regulatory network and applying this model to TGF-β-induced EMT they found that EMT is a sequential two-step program, in which an epithelial cell first is converted to a partial EMT (p-EMT) state and then to the mesenchymal state, depending on the strength and duration of TGF-β stimulation. Mechanistically, the process is governed by coupled reversible and irreversible bistable switches. While the Snail1/miR-34 double-negative feedback loop regulates the initiation of EMT and is reversible, the Zeb/miR-200 feedback loop controls the establishment of the mesenchymal state and is irreversible. Of note, irreversibility of the second switch and maintenance of EMT is assured by an aTGF-β/miR-200 feedback loop [31]. Likewise, in mammary MCF10A cells, stabilization of the mesenchymal state also involves a novel autocrine mechanism composed of a circuit with TGF-β, miR-200, and Snail1 [32].

Hepatic progenitor cells usually expand in chronic liver injury and contribute to liver regeneration by differentiating into hepatocytes and cholangiocytes. During this process, they acquire p-EMT states that are maintained by aTGF-β, activin A and Smad signaling [33]. Likewise, during in vitro differentiation to the hepatic lineage, human embryonic stem cells undergo a sequential EMT-mesenchymal-epithelial transition (MET) process with an obligatory intermediate mesenchymal phase. Remarkably, aTGFβ signaling mediates a synchronous EMT that accompanies activin A-induced formation of definitive endoderm and is followed by a MET process [34].

Autocrine TGF-β signaling also plays an essential role in the retention of stemness of glioma-initiating cells (GICs). Treatment of GICs with TGF-β signaling inhibitors promoted their differentiation, resulting in decreased tumorigenicity as evidenced by lower lethal potency in intracranial transplantation assays. In additional experiments, the authors identified an essential pathway for GICs involving aTGF-β, Sox4 and Sox2, whose disruption could be a therapeutic strategy against gliomas [35].

Yang and coworkers, by modulating TβRII levels, found that aTGF-β signaling transcriptionally targets human telomerase reverse transcriptase (hTERT) for inhibition of telomerase activity. Restoration of aTGF-β activity in TβRII-deficient HCT116 cells after re-expression of TβRII led to a reduction of hTERT mRNA levels and telomerase activity, whereas suppression of aTGF-β signaling in MCF-7 cells by dnTβRII had the reverse effect [36].

## 3. Autocrine TGF-β in the Regulation of Specific Proteins

### 3.1. Receptor Tyrosine Kinases, Adapter Proteins, E3 Ligases, and Small GTPases

Up-regulation and activation of the receptor tyrosine kinase, Axl, in EMT-transformed hepatoma cells caused phosphorylation of Smad3 in its linker region, resulting in the induction of PAI-1, matrix metalloproteinase-9 (MMP-9), and Snail as well as TGF-β1 secretion in mesenchymal HCC cells. Consistent with this, high Axl expression in HCC patient samples correlated with elevated vessel invasion of HCC cells, higher risk of tumor recurrence after liver transplantation, and lower patient survival [37]. Autocrine TGF-β also increased the endogenous levels of the adapter protein Crk and collaborates with Crk to form a positive feedback loop to facilitate EMT in A549 human lung adenocarcinoma cells through differential regulation of Rac1/Snail and RhoA/Slug [38].

TGF-β drives EMT through TβRI, which initiates both Smad-dependent and independent reprogramming of gene expression. TβRI is a dual-specificity kinase, which has tyrosine kinase activity and can activate the ERK mitogen-activated protein kinase (MAPK) pathway through recruitment and tyrosine phosphorylation of the adapter protein, ShcA [17]. Interestingly, ShcA protects the epithelial integrity of nontransformed cells against EMT and EMT-associated events by competing with Smad3 for TGF-β receptor binding and blocking aTGF-β/Smad signaling and target gene expression [18].

The tumor suppressor PTEN (phosphatase and tensin homolog on chromosome ten) is a substrate for XIAP E3 ubiquitin-protein-ligase activity, which decreases PTEN protein levels. In turn, XIAP gene expression and function is positively regulated by all three TGF-β isoforms in a Smad and NFκB-dependent manner. Moreover, its constitutive expression in endometrial and cervical carcinoma cells depends on aTGF-β signaling, together implicating aTGF-β/Smad signaling in XIAP-mediated downregulation of PTEN [39].

Recent data from the author’s group have shown that RAC1b, a splice isoform of the human *RAC1* gene, is a powerful inducer of aTGF-β1 production and signaling in pancreatic ductal adenocarcinoma (PDAC) and BC-derived cells [40] and is involved in the inhibition of EMT and cell migration. We have recently identified a novel tumor-suppressive pathway, in which RAC1b-driven aTGF-β1 induces Smad3 expression in an exogenous TGF-β-independent manner. Smad3 was subsequently identified as a potent suppressor of EMT and cell migration via its ability to maintain E-cadherin expression and to induce expression of the small proteoglycan biglycan, an extracellular TGF-β binding protein and inhibitor of TGF-β signaling [41]. Of note, in renal tubular epithelial cells, exposure to exogenous TGF-β1 for longer time periods (5–7 days) decreased Smad3 levels, which paralleled the EMT process. Down-regulation of Smad3 could be part of a feedback loop controlling TGF-β signaling in a cell phenotype-specific manner [42]. However, based on our results, a reduction in Smad3 abundance may also remove a barrier to induction of EMT and invasive activities [41], mediating relief from the tumor-suppressive effects of aTGF-β1 and RAC1b. In addition, we observed that in the same cells, aTGF-β1 promoted proliferation and therefore antagonized the action of exogenous (recombinant human) TGF-β1 on these cells [43]. Very recent results with cells, in which the endogenous *TGFB1* gene had been silenced, indicate that aTGF-β1 can even act an endogenous inhibitor of exogenous, recombinant human (rh)TGF-β1 [44].

### 3.2. Transcription Factors

Overexpression in mammary epithelial cells of Krüppel-like zinc finger protein ZNF217, a transcription factor (TF) and candidate oncogene in BC, stimulated EMT, migration and invasion in vitro and promoted the development of lung or node metastases in mice in vivo. TGF-β/Smad signaling was identified as a major driver of ZNF217-induced EMT and a TGF-β autocrine loop maintained by ZNF217-mediated up-regulation of *TGFB2* or *TGFB3* sustained activation of the TGF-β pathway in ZNF217-overexpressing BC cells [45]. Likewise, expression of the T-box TF, Brachyury, is enhanced during TGF-β1-induced EMT in various human cancer cell lines. Brachyury over-expression promoted up-regulation of TGF-β1 through activation of the *TGFB1* promoter, while inhibition of TGF-β1 signaling decreased the expression of this TF, eventually resulting in the establishment of a positive feedback loop between Brachyury and aTGF-β1 in mesenchymal-like tumor cells [46]. Another TF, forkhead box F2 (FOXF2), is over-expressed in basal-like breast cancer (BLBC) cells and suppressed EMT and malignancy of these cells. FOXF2 repressed TGF-β/Smad signaling in BLBC cells, while, in turn, TGF-β down-regulated FOXF2 expression through induction of miR-182-5p. FOXF2-deficient BLBC cells converted to a CAF-like phenotype and showed a propensity for metastases formation in visceral organs by increasing aTGF-β signaling and by making neighboring cells more aggressive through enhancing signaling by paracrine TGF-β [47].

Glioblastoma multiforme (GBM) is characterized by aberrantly high levels of TGF-β2, a result of TGF-β2 auto-induction through a positive autocrine feedback loop. The TF, cAMP-responsive element-binding protein 1 (CREB1), was identified as the main perpetuator of this circuit. CREB1 binding to the *TGFB2* promoter in cooperation with Smad3 was required for TGF-β2 to activate transcription. Since in patient-derived in vivo models of glioblastoma, CREB1 levels have been found to determine the expression of *TGFB2*, CREB1 has been proposed a biomarker to stratify GBM patients for anti-TGF-β treatments and eventually as a therapeutic target in anti-TGF-β therapies [48].

ATF3, an adaptive-response gene, is induced by various stromal signals, i.e., TGF-β, in MCF10CA1a BC cells and is crucial for TGF-β-induced up-regulation of Snail, Slug and Twist, and enhancement of cell motility. Since ATF3 also up-regulates the *TGFB* gene(s), it forms a positive feedback loop to drive TGF-β signaling. Not surprisingly, therefore, ectopic expression of ATF3 led to EMT and features associated with BC-initiating cells [49].

Paraspeckle component 1 (PSPC1) is up-regulated and associated with poor survival in cancer patients. It enhances EMT and stem cell phenotypes in multiple cell types as well as metastasis in mouse models of spontaneously arising cancers. PSPC1 is a master activator of EMT-TFs, increases TGF-β1 secretion to amplify aTGF-β1 signaling, and controls the pro-metastatic switch of TGF-β1 from a tumor suppressor to a tumor promoter [50].

NCI-H358 non-small cell lung cancer (NSCLC) cells engineered to express Snail, Zeb1 or activated TGF-β exhibited aTGFβ expression and phenotypic changes consistent with EMT and a shift to a more invasive phenotype. Although Snail and Zeb1 were sufficient to induce EMT in these cells, aTGFβ induced a more complete EMT phenotype [51].

### 3.3. MicroRNAs

The aTGF-β/Zeb/miR-200 signaling network regulates plasticity between epithelial and mesenchymal states in invasive ductal carcinomas, including those of the human breast. Both the induction and maintenance of a stable mesenchymal phenotype requires the establishment of aTGF-β signaling to drive sustained ZEB expression, while prolonged aTGF-β signaling induces reversible DNA methylation of the miR-200 loci with corresponding changes in miR-200 levels [52]. Rateitschak and colleagues have developed kinetic models to describe how aTGF-β signaling induces and maintains an EMT by up-regulating ZEB1 and ZEB2, which in turn represses the expression of miR-200b/c family members. When combined with data from patient-derived tumor cells, their algorithms can predict the minimal amount of an inhibitor required to induce a MET [53]. In A549 cells, TGF-β1 cooperates with yet another miR, hsa-miR-21, in the induction of EMT. Intriguingly, TGF-β1 was found to induce hsa-miR-21 expression and both are involved in autocrine and paracrine circuits that regulate the EMT status of lung cancer cells [54]. In NSCLC tissues, expression of another miR, miR-124, is significantly impaired and is associated with metastasis. Restoring miR-124 expression in NSCLC cells reduced migration, invasion and metastasis. Smad4 was identified as a novel target gene of miR-124, suggesting that a feedback loop between miR-124 and the TGF-β pathway may play a crucial role in NSCLC metastasis [55]. In lung adenocarcinoma H1299 xenograft assays, stable expression of miR-206 suppressed both tumor growth and metastasis in mice. Profiling of xenograft tumors revealed a network of genes involved in TGF-β signaling that were regulated by miR-206, i.e., *TGFB1*, direct transcriptional targets of Smad3, and components of the ECM involved in TGF-β activation, i.e., thrombospondin-1. Hence, miR-206 can suppress tumor progression and metastasis by limiting the production of aTGF-β [56]. In BC, decreased miR-206 expression is associated with advanced clinical stage and lymph node metastasis, and miR-206 overexpression in ER-positive cell lines markedly impaired EMT, migration, invasion, and inhibited *TGFB1* transcription and aTGF-β1 production [57]. Stabilization of *TGFB1* mRNA, its translation and expression, TGF-β1 dimer formation and aTGF-β1-induced EMT was also promoted by N6-methyladenosine, the most abundant modification on eukaryotic mRNA [58].

### 3.4. Membrane Proteins and Integrins

Induction of EMT in mouse mammary epithelial EpH4 cells by an inducible c-fos estrogen receptor (FosER) oncoprotein involves production of aTGF-β, and inhibition of TGF-β signaling in the mesenchymal FosER cells caused a MET. Additional results demonstrated that increased LEF/TCF/β-catenin signaling resulting from a loss of E-cadherin cooperated with aTGF-β signaling in maintaining an undifferentiated mesenchymal phenotype [59]. EMT and invasion in MCF-7 cells was also promoted by physical interactions between platelets and tumor cells via direct contacting of surface integrin α2β1. This integrin activated the Wnt/β-catenin and TGF-β1/Smad3 pathways via aTGF-β1 production, which together promoted the expression of EMT proteins [60]. In HCC, sphingosine-1-phosphate (S1P) induced EMT via activation of PI3K/AKT signaling, which triggered heparanase, leading to increased expression and activity of MMP-7 and shedding and suppression of SDC1. The loss of SDC1, in turn, caused an increase in TGF-β1 production, which can convert HCC cells to a mesenchymal phenotype via establishing an MMP-7/SDC1/aTGF-β1 autocrine loop [61]. Additionally in HCC cells, silencing of the oncogene, mucin1 (MUC1), decreased TGF-β signaling, while its overexpression enhanced the levels of Smad3 phosphorylated in its linker region, and of MMP-9. MUC1 also stimulated endogenous *TGFB1* transcription, protein production/secretion and cell migration, which were markedly inhibited by either TβR inhibitor or silencing of *TGFB1* [62]. Myoferlin, a protein involved in plasma membrane function and repair, is overexpressed in several invasive cancer cell lines, and in MDA-MB-231 BC cells promoted EMT, migration and invasion by enhancing endogenous *TGFB1* transcription and TGF-β1 protein secretion. This study identified regulation of aTGF-β signaling as a novel mechanism by which myoferlin regulates cellular phenotype and invasive capacity of human BC cells [63].

### 3.5. Secreted Proteins and Enzymes Regulating Growth Factor Bioavailability

Secreted hominoid-specific oncogene (SHON), a secreted protein expressed in all human cancer cell lines tested, has oncogenic potential for human BC cells. Its ectopic overexpression in immortalized HMECs was sufficient for these cells to acquire mesenchymal traits and epithelial stem cell properties, and to enhance cell migration and invasion. Intriguingly, SHON contributed to EMT induction by activating aTGF-β signaling, while SHON itself was induced by TGF-β, suggesting that a SHON-TGF-β-SHON positive feedback loop controls EMT in BC progression [64].

By activating aTGF-β3 signaling collagen I induces EMT in NSCLC cells, which is prevented by inhibitors of PI3K and ERK signaling, indicating that these MAPKs promote transcription of TGF-β3 mRNA [65]. Auto- and paracrine-mediated induction of EMT was also triggered by MT1-MMP-mediated activation of aTGF-β signaling. While MT1-MMP failed to affect total TGF-β levels, its catalytic activity increased the availability of bioactive TGF-β, enabling MT1-MMP-expressing cells to induce EMT and eventually tumor cell invasion in nearby cells [66].

An autocrine loop of TGF-β involving an increase in the expression of EGFR ligands confers resistance to apoptosis in hepatocytes after these have undergone an EMT [67]. Similarly, in MMTV-Neu transgenic mice, TGF-β enhanced metastasis of mammary tumors, induced EMT by establishing an autocrine platelet-derived growth factor (PDGF)/PDGFR loop, and elevated PDGFR signaling [68]. The lack, or inhibition, of heparanase (HPSE), an endo-β-D-glucuronidase that cleaves heparan-sulfate to regulate the bioavailability of fibroblast growth factor (FGF)-2 and TGF-β, prevented the increased synthesis of TGF-β by tubular cells in response to pro-fibrotic stimuli, thereby interfering with a self-sustaining aTGF-β loop. Hence, HPSE is needed for pathological TGF-β overexpression in response to pro-fibrotic factors and for TGF-β-induced EMT [69].

TGF-β is synthesized as a precursor molecule and proteolytically processed to the mature form by the proprotein convertases subtilisin/kexin furin, which is highly expressed in human GICs. Furin cleaves and activates the proforms of TGF-β1 and TGF-β2, while, conversely, furin expression and activity levels were stimulated byTGF-β2 in a TβRI and ERK-dependent manner. This study has thus identified a self-sustaining loop for high TGF-β and furin activity in GICs [70].

### 3.6. Environmental and Metabolic Factors

The TME and its physical properties are known to impact tumor cell phenotype and to foster tumor progression [6]. For instance, in gastric cancer, hypoxia was associated with distant metastasis and in vitro stimulated EMT of gastric cancer cells via expression of TGF-β1 mRNA and aTGF-β/TβR signaling [71]. TGF-β1 derived from hypoxic bone marrow stroma promoted the growth of human BC stem cells as mammospheres, and Slug/β-catenin-dependent activation of DNA damage signaling triggered by the hypoxic microenvironment sustained their proinflammatory phenotype [72].

Acidosis is a hallmark feature of the TME and correlates with fluctuations in the cancer cells’ metabolic activities and with disease progression. An acidic pH facilitated signaling through aTGF-β2, which promoted the formation of lipid droplets. These lipid droplets served as energy stores to support resistance to apoptosis and invasive activities of the tumor cells. In various types of cancer cells, acidosis-induced TGF-β2 signaling enhanced both p-EMT and fatty acid metabolism, leading the authors to speculate that inhibiting aTGF-β2 signaling and, as a consequence, fatty acid mobilization from lipid droplets, could be a strategy to prevent distant metastatic spreading [73]. Interestingly, exposure of mesothelioma cells to acidosis promoted TGF-β2 secretion, which, in turn, led to lipid droplet accumulation and profound metabolic rewiring in dendritic cells (DCs). The acidic mesothelioma milieu drove DC dysfunction and altered T cell response through TGF-β2-dependent mechanisms [74].

In bladder cancer cells, starvation increased the expression of TGF-β1 and phosphorylated Smad3, and enhanced EMT-mediated invasion and migration, while inducing autophagy. Moreover, autophagy and TGF-β1 can form a positive feedback loop to synergistically promote invasion and migration [75]. In human glioma cells, autophagy activated TGF-β2, while conversely, TGF-β2 could initiate autophagy via Smad and non-Smad pathways. The autocrine loop between autophagy and TGF-β2 promoted EMT, metabolic conversion and glioma cell invasion [76].

Autocrine TGF-β is also involved in glucose metabolism. For example, tumor cell-derived angiopoietin 2 (ANGPTL2) has a role in establishing a preference for glycolytic metabolism. ANGPTL2 signaling in lung cancer cells through integrin α5β1 enhanced GLUT3 expression by increasing aTGF-β signaling and expression of ZEB1. The aTGF-β-ZEB1-GLUT3 axis accelerated activities associated with a glycolytic metabolism [77]. GRP78 (glucose-regulated protein of 78kD) belongs to the heat shock protein 70 family and its up-regulation in response to physiological or environmental stressors is positively associated with tumor initiation and progression. GRP78 overexpression in CRC cells facilitated TGF-β1 expression and secretion as well as Smad signaling to regulate EMT [78]. Analysis of the role of the glucose-transforming polyol pathway (PP) in TGF-β-dependent EMT provided novel mechanistic insights into the metabolic control of cancer cell differentiation. Expression of the gene encoding aldo-keto-reductase-1-member-B1 (*AKR1B1*) was found to be strongly associated with EMT and silencing *AKR1B1* reverted EMT and repressed TGF-β signature genes. In control cells, hyperglycemia promoted EMT through aTGF-β stimulation, however, the PP-deficient cells failed to respond to high glucose-induced EMT. These data establish a molecular link between PP, glucose metabolism, aTGF-β signaling and cancer cell EMT [79]. Another variation of this theme discovered by Rahn and colleagues might explain how type 2 diabetes mellitus (T2DM), which is associated with hyperglycemia and represents a risk factor for the development of PDAC, facilitates pancreatic tumorigenesis. This group addressed the question if hyperglycemia can promote EMT and CSC features in premalignant pancreatic ductal epithelial cells (PDEC). In premalignant H6c7-kras cells, high glucose increased secretion of aTGF-β1 as well as TGF-β1 signaling activity, and in a TGF-β1-dependent manner reduced E-cadherin and increased stem cell marker expression. Hence, hyperglycemia promoted the acquisition of mesenchymal and CSC properties in PDEC by activating aTGF-β signaling [80].

Lipoxin A4 (LXA4), one of the metabolites derived from arachidonic acid, has recently been reported to exhibit anti-cancer effects and in PDAC patients, a low lipoxin effect score (LES) was associated with aggressive metastatic potential. LXA4 strongly suppressed the expression and signaling activity of aTGF-β1 and reversed mesenchymal phenotypes and invasive capacity [81]. Lastly, overexpression of DEPTOR, an mTOR-interacting protein, whose expression is negatively regulated by mTORC1 and mTORC2, promoted the invasion and metastasis of HCC cells in vitro and in vivo. DEPTOR induced an EMT and metastasis via up-regulation of Snail, which was due, in turn, to activation of aTGF-β1-Smad3/4 signaling, possibly through feedback inhibition of mTOR [82].

### 3.7. Paracrine Interactions between Non-Cancerous Cells and Cancer Cells

In a co-implantation BC xenograft model, resident human mammary fibroblasts progressively converted into CAFs during the course of tumor progression. These cells increasingly acquired two autocrine signaling loops, mediated by TGF-β and stromal cell-derived factor 1 (SDF-1) cytokines, which both act in autostimulatory and cross-communicating fashions to initiate and maintain the differentiation of fibroblasts into myofibroblasts and the concurrent tumor-promoting phenotype [83]. The molecular mechanism underlying sustainment of the active status of CAFs is largely unknown. In CAFs, DNA (cytosine-5-)-methyltransferase 3 beta (DNMT3B) was not only a target of miR-200b, miR-200c and miR-221, but was able to induce DNA methylation of the miR-200 promoters. Suppression of miR-200 levels by DNMT3B established CAF activation from normal fibroblasts with miR-200/miR-221/DNMT3B signaling sustaining aTGF-β1 signaling and, as a consequence, the active CAF status. This study showed that the TGF-β1/miR-200/miR-221/DNMT3B regulatory loop was crucially involved in maintaining CAF status as well as CAF function in promoting BC malignancy [84]. Additionally in BC, MCF-7 cells after prolonged co-culture with human adipose-derived mesenchymal stem cells MSCs (hAD-MSCs) underwent EMT and established a stable mesenchymal phenotype. By targeting the ZEB/miR-200 regulatory loop, paracrine TGF-β1 secreted by hAD-MSCs regulated the establishment of EMT in MCF-7 cells, while maintaining of a stable mesenchymal state in MCF-7 cells required aTGF-β signaling to drive and sustain ZEB expression [85].

### 3.8. Cooperation with Oncogenes and Tumor Suppressor Genes

Cells transformed by oncogenic Ras often lose their inhibitory responses and instead show an increase in malignancy following TGF-β treatment, a phenomenon termed mitogenic conversion. Immortalized murine hepatocytes transformed with oncogenic HRas rapidly converted to a spindle-shaped morphology upon treatment with TGF-β1, which no longer inhibited proliferation. The fibroblastoid cells secreted high levels of TGF-β1, suggesting aTGF-β signaling and grew to severely vascularized tumors in vivo. In collaboration with activated HRas TGF-β1 thus promoted late malignant events in hepatocytes [86]. Likewise, prostate carcinomas with Ras/MAPK pathway activation might have a selective growth advantage through aTGF-β1 production [87].

In the canine kidney-derived epithelial cell line, MDCK, synergism between activation of the Raf/MAPK pathway and the resulting production of aTGF-β triggered an EMT during which these cells became refractory to TGF-β-induced cell cycle arrest and apoptosis. Mesenchymal phenotype conversion was accompanied by gradual down-regulation of the expression of Smad3, an event also critical for induction of a migratory phenotype [41]. Re-expression of Smad3 expression restored the TGF-β-induced cell cycle arrest without reverting the cells to an epithelial phenotype. These data attribute to Smad3 a crucial role in the control of cell proliferation by TGF-β, which is lost following an EMT [88].

In a combined in vitro/in vivo carcinogenesis model with HRas-transformed mammary epithelial cells (EpRas), HRas cooperated with TGF-β to cause EMT. This EMT required continuous TβR and oncogenic Ras signaling and was stabilized by aTGF-β production and an aTGF-β autocrine loop. DnTβRII blocked TβR signaling, prevented EMT and delayed tumor formation. Hence, both EMT and metastasis required synergism between aTGF-β/TβR and Raf/MAPK signaling [89]. However, aTGF-β can also block transformation. In a series of isogenic HMECs representing a full spectrum of BLBC, Lin and colleagues identified candidate genes that mediate the escape from OIS and malignant transformation in BLBC. They found that aTGF-β signaling is an integral part of a cellular anti-transformation network due to the ability of aTGF-β to suppress the expression of genes, including p21^CIP1^-regulated genes that mediate escape from OIS. Intriguingly, abrogation of aTGF-β signaling by dnTβRII promoted malignant progression of HRas^GV12^-transformed HMECs both in vitro and in vivo. These findings violate the current dogma that TGF-β inhibits breast carcinogenesis merely by its growth-arresting function and provide novel insights into the mechanism of the cross-talk between aTGF-β and oncogenic Ras signaling in BLBC development and progression [12].

In a more recent study with 3-D cultures of Kras^G12D^-expressing mouse pancreatic epithelial cells, it was demonstrated that while exposure to exogenous TGF-β induced growth arrest of the Kras^G12D^ cells, its subsequent removal allowed the cells to enter a hyper-proliferative, partially mesenchymal (PM), and progenitor-like state. This state was highly stable and was maintained by aTGF-β signaling. In vivo, PM cells resembled human precursor lesions (PanINs), suggesting that they had attained increased oncogenic potential, in agreement with shared molecular and phenotypic features of the aggressive quasi-mesenchymal/squamous subtype of human PDAC. In this system, transient TGF-β exposure was sufficient to induce the acquisition of PDAC-associated phenotypes in pre-neoplastic Kras^G12D^ cells, the maintenance of which required aTGF-β signaling. This study provided novel molecular insight into the complex role of TGF-β in tumorigenesis [90].

In immortalized human diploid fibroblasts, oncogenic rewiring by transduction of HRas^GV12^ instigated regulation of RhoA-ROCK signaling through an aTGF-β1-TβRI pathway. Moreover, TβRI-mediated activation of RhoA was required for efficient HRas^GV12^ and BRAF^V600E^-induced transformation and HRas^GV12^-mediated anchorage-independent growth, identifying a novel pro-oncogenic activity of TGF-β [91].

Variants within the binding sites of miRs located in the 3′-untranslated region (3′-UTR) of cancer-driving genes are a new class of germ-line mutations, which are increasingly recognized as suitable biomarkers in human cancer. A let-7-binding site mutation in the 3’UTR of *KRAS* is among the first mutations discovered in this class. Its occurrence is associated with increased cancer risk and is predictive of drug response and elevated TGF-β and immunosuppression in cancer patients. KRAS-variant normal breast epithelial cells exhibit a mesenchymal phenotype, which appears to be due to numerous molecular changes and pathway alterations, including elevated aTGF-β signaling, resulting in ZEB and SNAIL up-regulation [92].

Another study identified R-Ras2 as a critical regulator of TGF-β signaling in vivo. The authors genetically deleted Nf1 (a RasGAP protein) to activate all Ras proteins in vivo followed by examination of mice double-deficient in a specific Ras protein and Nf1 to assess its requirement in the generation of TGF-β-dependent neurofibromas that arise in Nf1-null mice. In animals lacking the Ras-related protein R-Ras2/TC21 (in addition to Nf1) the formation of neurofibromas was delayed, while that of sarcomas and brain tumors was accelerated. Of note, loss of R-Ras2 was associated with increased expression of TGF-β in Nf1-deficient Schwann cell precursors, blockade of a Nf1/TβRII/AKT-dependent autocrine loop in premalignant precursor cells, and a decline in the number of these cells. In malignant tumors of peripheral nerve sheaths, an increase in TGF-β ligands and a loss of TβRII was observed, together pointing to R-Ras2 as a critical regulator of TGF-β signaling in vivo [93].

Sustained activation of Raf in MDCK cells was able to induce an EMT and invasive growth, and this was dependent on an autocrine loop involving TGF-β, whose secretion was induced by Raf. Activation of Raf led to inhibition of the ability of TGF-β to induce apoptosis but not growth arrest and allowed the cells to respond to TGF-β with increased invasiveness. Like Ras, the Raf-MAPK pathway thus synergizes with TGF-β in promoting malignancy but unlike Ras does not directly impair TGF-β/Smad-induced growth inhibition [94].

The activating mutation, BRAF^V600E^, is a frequent genetic event in papillary thyroid carcinomas (PTCs) that predicts poor prognosis, leading to loss of sodium/iodide symporter (NIS) expression and subsequent radio-iodide-refractory metastatic disease. BRAF induces secretion of functional TGF-β and blocks TGF-β/Smad signaling at multiple levels. Not surprisingly, therefore, TGF-β and other key components of TGF-β signaling are overexpressed in human PTC. Moreover, secreted TGF-β and high TGF-β/Smad activity cooperate with MEK-ERK signaling in BRAF-induced EMT, cell migration, invasion, and nodal metastasis. These data provide evidence that TGF-β plays a key role in promoting radioiodide resistance and tumor invasion during PTC progression [95].

SMAD4 is a central transducer of TGF-β’s growth-inhibitory effects but it can also block invasiveness by inducing a MET. Of note, in SW480 CRC cells, SMAD4 down-regulated endogenous TGF-β cytokines, suggesting that suppression of aTGF-β signaling represents one mechanism through which Smad4 interferes with EMT [96]. However, these observations are somehow ad odds with others showing that aTGF-β supported cancer cell invasion by stimulating secretion of urokinase-type plasminogen activator (uPA) in a Smad4-dependent manner [97].

The oncogenic potential of a point mutant of c-Myc (Myc^V394D^) that is selectively deficient in binding to Miz1 is greatly attenuated. Binding of Myc to Miz1 is required to antagonize TGF-β-induced growth arrest and senescence. Since T-cell lymphomas express high levels of TGF-β, they are poised to elicit an autocrine program of senescence upon Myc inactivation, demonstrating that TGF-β is a key factor in establishing oncogene addiction of lymphomas [98].

N-myc downstream-regulated gene 2 (NDRG2) has been studied for its anti-proliferative and anti-metastatic effects in various tumor cell types. In BC, NDRG2 expression in human tissues is negatively associated with lymph node metastasis and pTNM stage and positively with recurrence-free patient survival. NDRG2-overexpressing mouse 4T1 cells showed less Smad-dependent-transcription and lower levels of active aTGF-β, invasiveness in vitro and metastatic activity in vivo than control cells, suggesting that NDRG2 suppresses tumor metastasis by attenuating active aTGF-β production and signaling [99].

Epsin 3 (EPN3), an oncogene with prognostic and therapeutic relevance in BC, drives breast tumorigenesis by increasing endocytosis of E-cadherin, followed by the activation of a β-catenin/TCF4-dependent p-EMT and establishment of an aTGFβ-dependent autocrine loop that sustains EMT. EPN3-induced p-EMT and aTGFβ-dependent signaling are, therefore, crucial for conferring cellular plasticity and invasive behavior, and the transition from in situ to invasive BC [100].

Disabled-2 (Dab2) is a putative tumor suppressor, whose expression is down-regulated in various cancer types including BC. Decreased Dab2 expression in HMECs led to the appearance of a constitutive EMT phenotype and increased Ras/MAPK signaling. This facilitated the establishment of an aTGFβ signaling loop, concomitant with increased expression of TGF-β2. Loss of Dab2 expression may thus facilitate aTGFβ-stimulated EMT and metastasis [101].

Cells transformed by mutant HER2 are resistant to EGFR tyrosine kinase inhibitors and exhibit an attenuated response to the HER2 antibody, trastuzumab. HMECs expressing mutant HER2, or HRas^G12V^, expressed higher levels of TGF-β1 along with activated aTGF-β1 signaling through a mechanism involving Rac1 activation. Strikingly, inhibition of aTGF-β signaling with the TβRI inhibitor, LY2109761, reduced growth and invasiveness of cells expressing mutant HER2, providing a rationale for combining anti-EGFR with anti-TGF-β therapies [102].

### 3.9. Therapy-Associated aTGF-β Signaling

Ionizing radiation (IR), a well-established treatment in many human cancers, is known to induce EMT and migration in cancer cells. In A549 cells, IR triggered the synthesis and secretion of both aTGF-β1 and activin A as well as TGF-β/activin signaling activity and these responses were sensitive to SB431542, a pan-specific inhibitor of TβRI and the activin type I receptor, ALK4. However, only specific antibody-mediated neutralization of TGF-β1, or dn interference with TβRI or TβRII but not ALK4 function alleviated the IR-induced EMT and cell migration, proving that production of aTGF-β1 and subsequent activation of TGF-β but not activin signaling mediated the potentially hazardous effects of IR [103].

In human breast stromal fibroblasts in vitro and in breast tissue in vivo, IR provokes premature senescence. The senescent cells overexpress syndecan-1 (SDC1), a poor prognostic factor for cancer growth, which is the result of aTGF-β acting through the Smad pathway. In addition, highly invasive MDA-MB-231 cells also enhance SDC1 expression in both early-passage and senescent fibroblasts, via a paracrine action of TGF-β. Hence, both radiation-mediated premature senescence and invasive tumor cells, alone or in combination and in an autocrine/paracrine TGF-β-dependent manner, can enhance SDC1 expression in breast stromal fibroblasts [104].

In the MMTV/PyVmT transgenic model of metastatic BC, administration of IR or doxorubicin elevated blood levels of TGF-β1, and the number of circulating tumor cells and lung metastases. These radiation effects were abrogated in mice bearing tumors that lack TβRII, or by administration of a neutralizing pan-TGF-β antibody. Additionally, in the presence of this antibody, circulating PyVmT-expressing tumor cells failed to grow ex vivo, suggesting that aTGF-β is a survival factor for these cells and that the increase in metastases was due, at least in part, to a direct effect of TGF-β on the cancer cells [105].

In human GBM tissues, the oncogene MSH6, CXCR4 and TGF-β1 form a triangular feedback loop that accelerates gliomagenesis, proliferation, migration/invasion, EMT, stemness, angiogenesis and survival by regulating the STAT3/Slug and Smad2/3/ZEB2 signaling pathways. Photothermal therapy in GBM mediated by Cu_2_(OH)PO_4_@PAA + near infrared irradiation showed excellent therapeutic effects. Since these were likely caused by repression of the MSH6-CXCR4-TGF-β1 feedback loop and its downstream targets, this TGF-β1 involving circuit emerges as a novel and promising therapeutic target in GBM [106].

Therapeutic pressure is known to activate effective resistance mechanisms in tumors. Steins and colleagues set out to characterize these mechanisms in response to the currently used neoadjuvant treatments against esophageal adenocarcinoma (EAC), radiotherapy carboplatin and paclitaxel. Application of this chemoradiation regimen of high therapeutic pressure to primary patient-derived cultures generated a heterogeneous EMT response in EAC cells with EMT being initiated by the production and response to aTGF-β. Inhibition of TGF-β ligands effectively abolished chemoradiation-induced EMT, indicating that chemoradiation contributed to resistant metastatic disease in EAC patients by aTGF-β-dependent EMT induction. Monitoring serum levels of TGF-β during treatment could identify those patients at risk of developing metastatic disease, and others who likely benefit from anti-TGF-β therapy [107]. The above referenced studies all describe TGF-β induced in response to IR as a pro-metastatic signal and provide a rationale for combining (chemo)radiation therapies with TGF-β inhibitors.

PDAC is characterized by a desmoplastic stroma, the generation of which is orchestrated by pancreatic stellate cells (PSCs). While healthy PSCs are quiescent, these cells adopt a myofibroblast-like phenotype following activation during disease progression, secrete matrix proteins and TGF-β, and organise a mechanically stiff ECM. All trans-retinoic acid (ATRA), which induces PSC quiescence, blocks the ability of PSCs to release active TGF-β, which would otherwise act in an autocrine manner to maintain PSCs in an activated state and promote a tumor-favoring stiff ECM [108]. The most widely used antidiabetic drug, metformin, also suppresses desmoplasia in PDAC as well as cancer cell migration and invasion. Intriguingly, a study by Duan et al. found that metformin potently inhibited EMT, TGF‑β1 production and Smad2/3 phosphorylation in pancreatic cancer cells, leading to the identification of a novel antitumor-mechanism of metformin in PDAC, inhibition of aTGF‑β1/Smad signaling [109].

The impact of EGFR-mutant NSCLC precision therapy with EGFR tyrosine kinase inhibitors, such as erlotinib and gefitinib is limited by acquired resistance despite good initial responses but the underlying mechanisms remain elusive. Thiagarajan and colleagues have characterized a novel non-mutational early adaptive and MET-independent drug escape in EGFR-mutant lung tumor cells only days after therapy initiation. The phenotypes of cells that managed to escape drug treatment revealed a central role for aTGF-β2 in mediating cellular plasticity through profound cellular adaptive Omics reprogramming with a common mechanistic link to prosurvival mitochondrial priming. Cells exhibiting early adaptive drug escape were growth-arrested, metabolically quiescent, and showed features of enhanced EMT and stem cell signaling. In addition, they exhibited global suppression of bioenergetics including reverse Warburg, and were susceptible to glutamine deprivation and TGF-β2 inhibition [110].

As for EGFR-mutant NSCLC, many patients with TNBC suffer recurrence of drug-resistant metastatic disease after an initial response to chemotherapy. Studies with TNBC cells suggest that chemotherapy-resistant populations of CSCs are responsible for the tumor relapse. When comparing RNA expression signatures in matched pairs of primary BC biopsies before and after chemotherapy, biopsies after chemotherapy displayed increased activity of genes associated with CSCs and TGF-β signaling, consistent with the ability of TGF-β to increase stem-like properties in human BC cells. In TNBC cell lines and mouse xenografts, paclitaxel increased aTGF-β signaling and enriched for CSCs, as indicated by mammosphere formation and CSC markers. The TβRI inhibitor LY2157299, a neutralizing TβRII antibody, or SMAD4 small interfering RNA all blocked paclitaxel-induced expansion of CSCs. Moreover, treatment of TNBC xenografts with LY2157299 prevented tumor relapse after paclitaxel treatment, suggesting that chemotherapy-induced aTGF-β signaling enhances tumor recurrence through expansion of CSCs [111].

Programmed cell death ligand 1 (PD-L1) expression in cancer cells has been shown to be a predictive factor for the therapeutic efficiency of cancer immunotherapy. Funaki et al. investigated the mechanism of PD-L1 expression during the EMT process in NSCLC cells. PD-L1 expression was up-regulated in A549 cells following induction with rhTGF-β1 or the chemotherapeutic drug carboplatin, and down-regulated by SB431542 or subsequent removal of rhTGF-β1, which resulted in reversal of EMT. Chemotreatment increased TGF-β secretion, aTGF-β signaling, and PD-L1 expression via aTGF-β induced EMT. These data suggest that PD-L1 expression is controlled by TGF-β-induced EMT and enhanced by chemotherapy via drug-induced TGF-β signaling [112]. The various therapeutic strategies involving aTGF-β signaling are summarized in Figure 2.

## 4. General Considerations and Caveats in Studying Autocrine TGF-β Signaling

While paracrine signaling—the communication via soluble factors between two different types of cells—is more common in normal physiology, autocrine signaling loops with positive or negative feedback are an important feature in cancer, where they drive cancer-associated processes such as proliferation, survival and migration/invasion, and enable context-dependent cell signaling. Here, the composition of the TME can critically affect the intracellular signaling dynamics triggered by extracellular stimuli. More specifically, the signaling response to a transient input is short-lived when most of the released ligand is lost to the cellular microenvironment, i.e., by diffusion or interaction with an extracellular ligand-binding factor, but prolonged or persistent in a cell that efficiently recaptures the endogenous ligand. Shvartsman and colleagues developed a mathematical model that accounts for all parts of an autocrine loop by showing that context-dependent signaling arises as a result of dynamic interactions between ligand release, transport, binding, and intracellular signaling [113].

Autocrine and paracrine signaling mechanisms are traditionally difficult to investigate due to the sub-micromolar concentrations involved and the limited technology to detect these. A microfluidic cell culture perfusion system has been developed that could control the removal of molecular factors secreted by cells into the culture supernatants [114] and Blagovic and colleagues have employed this system to continuously remove the secreted factors to downregulate diffusible signaling. Moreover, by comparing cell growth and differentiation in side-by-side chambers with or without added cell-secreted factors, they were able to discern the effects of diffusible signaling from artifacts such as shear, nutrient depletion or micro-system effects [115]. Using this device, a comparison between minimal chemically-defined medium and medium fully supplemented with cell-secreted factors and in the absence or presence of added TGF-β signaling inhibitors, could be set up to determine the relative contribution of TGF-β and non-TGF-β secreted factors in promoting a specific cellular response in a specific cell type.

The studies discussed above impressively prove the crucial role of endogenous TGF-β and its eventual autocrine/paracrine modes of action in all cardinal processes driving tumor initiation and progression. The various TGF-β isoforms either operate on their own due to their ability to auto-induce their own expression or that of their receptors, or they cooperate with upstream activators or downstream effectors to either form self-perpetuating autocrine positive feedback/feedforward loops for signal amplification, or classical simple or multiple negative feedback loops for fine-tuning and stabilization of distinct cellular phenotypes. An example for the former type is TGF-β2 in GBM, and of the latter the well-known double-negative feedback loops of TGF-β with Snail/miR-34 or Zeb/miR-200, which operate in the induction and maintenance, respectively, of various EMT states along the spectrum from completely epithelial to completely mesenchymal [5].

In order to discern the autocrine mode of action of endogenously produced TGF-β, researchers employed a variety of different experimental strategies to either abrogate or enhance signaling. The majority of studies involved either silencing of one of the *TGFB* genes [62], antibody-mediated neutralization of secreted TGF-βs [103,105,107] or ectopic expression of active forms of TGF-β1 [32,33], naturally TβRII-deficient cell lines [19,36] *TGFBR2* knockout mice [105], dn inhibition of TβRII [12,21,26,29,36,103], antibody-mediated neutralization [111] or reconstitution of TβRII [4,36] or TβRIII [20], soluble Fc:fusion proteins of TβRII [22,26] or TβRIII [20], dn [102] or pharmacological inhibition of the TβRI kinase [19,23,34,62,102,103,111,112], ectopic expression of a kinase-active TβRI mutant [25], or dn inhibition or silencing of Smad2 [24] or Smad4 [111], respectively. However, unless combined with approaches that target the ligand(s) directly, deriving conclusions from the non-ligand-based strategies alone is not feasible, for the following reasons:

(1) Altering Smad expression does not allow for identification of either the receptor or the ligand, since Smads 2, 3 and 4 also transmit signals of ALK4 and ALK7, which are both activated by activins and nodal [116]. Moreover, they share some functional effects in common with the TGF-βs and eventually are produced by the same cells in response to chemo- or radiotherapy, such as activin A [103]. Only a few studies have attempted to rule out the participation of non-TGF-β ligands as drivers of autocrine signaling, i.e., activin A in EMT induction by IR in A549 cells [103]. This is particularly important when using small-molecule inhibitors of TβRI as most of them also target ALK4 and ALK7 for inhibition.

(2) Altering the expression or function of TβRI, TβRII, or TβRIII does not permit safe identification of the ligand actual responsible for their activation. This may be any of the three different TGF-β isoforms, another member of the superfamily of TGF-β ligands or even a completely unrelated factor. For instance, SCUBE3 (signal peptide-CUB-EGF-like domain-containing protein 3), a secreted glycoprotein promoting lung cancer invasiveness, also binds to TβRII and TβRI, activates TGF-β/Smad2/3 signaling, up-regulates the expression of target genes such as *TGFB1*, and triggers EMT [117]. SCUBE3 could thus serve as an endogenous autocrine and paracrine ligand of TβRII to regulate EMT and cancer progression. Likewise, two other members of the TGF-β superfamily of ligands, myostatin and GDF11, utilize TβRI/ALK5 [116]. Moreover, with respect to TβRII, it is not entirely clear if this receptor is needed for TGF-β in order to execute autocrine-mediated functions as aTGF-β has been reported to be able to induce gene expression and invasion in CRCs that lack TβRII as a result of microsatellite instability [19].

(3) Manipulations of TβR or Smad expression/function are expected to also alter the response to exogenous TGF-βs, thus precluding accurate assessments of how exogenous TGF-β1 interacts with aTGF-β1. This is a serious issue when trying to elucidate the effects of aTGF-β1 in vitro in cell culture experiments as most studies were performed with cells cultured in medium with 10% fetal bovine serum (FBS), which usually contains high concentrations of latent or bioactive TGF-β1 [44]. This non-endogenously produced TGF-β may activate the receptors even when the endogenous TGF-β genes have been silenced and may result in misinterpretation of data. To avoid this problem, experiments need to be performed under serum-free conditions and/or include neutralization of TGF-β with a pan-specific antibody in the culture supernatants, which would also neutralize the TGF-β(s) contained in the serum supplement and prevent them from binding to the receptors. However, only very few investigators have carried out their experiments under serum-free conditions [23]. Finally, the exposure of cells’ to exogenous TGF-β may alter the cells responsiveness to aTGF-β1, or vice versa, due to regulatory events at the receptor level, i.e., ligand-induced receptor internalization and desensitization [13]. In fact, we have preliminary data to indicate that endogenously produced/aTGF-β1 can decrease the cells’ sensitivity to stimulation with rhTGF-β1 with respect to gene regulatory as well as growth-inhibitory and invasive activities [44]. These caveats will hopefully raise awareness in those investigators, who intend to study autocrine effects of TGF-β or other growth factors and provide a guideline for a rational design of meaningful experiments.

## Figures and Tables

**Figure 1 ijms-22-00977-f001:**
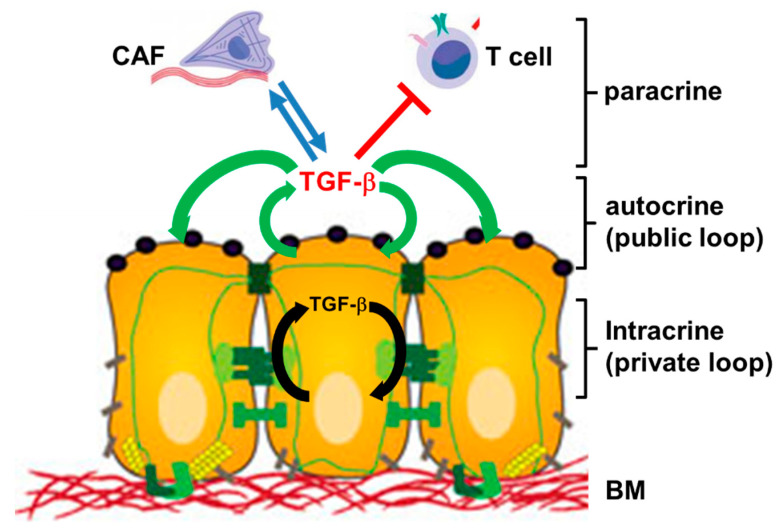
Cartoon illustrating the principles of autocrine, intracrine and paracrine TGF-β signaling. Autocrine interactions are indicated by green arrows, intracrine interactions by black arrows, and paracrine interactions by blue arrows (stimulatory) or red lines (inhibitory). Paracrine interactions can be unidirectional or bidirectional/reciprocal. CAF, cancer-associated fibroblast; BM, basement membrane.

**Figure 2 ijms-22-00977-f002:**
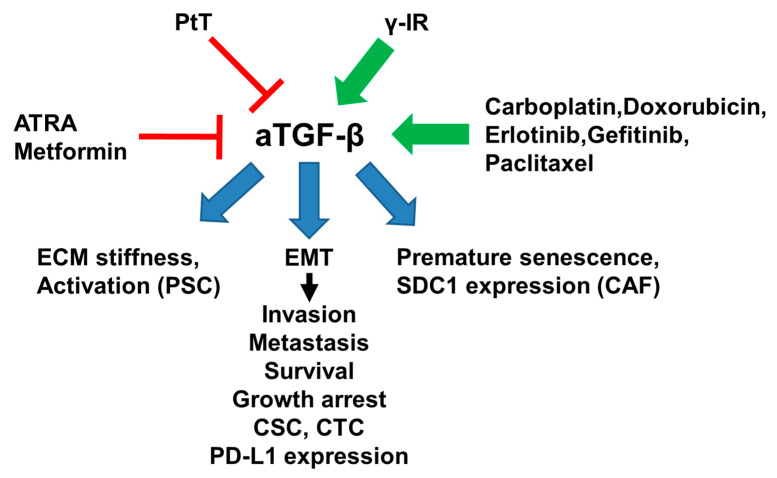
Cartoon illustrating various therapeutic interventions that converge on aTGF-β signaling. ATRA, all trans-retinoic acid; CSC, cancer stem cell; CTC, circulating tumor cell; PtT, photothermal therapy. Red lines denote inhibition, green arrows indicate induction/activation, and blue arrows point to the cellular processes targeted by aTGF-β. For details see text.

## Data Availability

Not applicable.

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
