# Peer review of "Autocrine TGF-β in Cancer: Review of the Literature and Caveats in Experimental Analysis"

_ijms, 2021, doi:10.3390/ijms22020977_

Round 1
Reviewer 1 Report
The review manuscript drafted by Hendrik Ungefroren elaborates in detail the autocrine effect of TGFb in the cancer setting. The paper is very well structured and written. There are only two minor pints that the author needs to apply before it goes to publication:
1) It is absolutely necessary to draw one or more cartoons to make it easier for the audience to read and follow the paper.
2) As stated in the introduction, tgfb has a great impact on immune cells and CAF. It would be great and more explanatory if the author could dedicate a section to this as well.
Thanks.
Author Response
Dear Editor, dear Arielia,
Please find enclosed the revised version of our manuscript (Manuscript ID: ijms-1073424). We thank the reviewers for their enthusiastic comments and have done our best to incorporate their suggestions in the revised version (highlighted in “track changes” mode). We believe that this has considerably improved its quality. We are looking forward to its final acceptance in the IJMS.
With warm regards,
Hendrik Ungefroren
Reviewer 1
The review manuscript drafted by Hendrik Ungefroren elaborates in detail the autocrine effect of TGFb in the cancer setting. The paper is very well structured and written. There are only two minor pints that the author needs to apply before it goes to publication:
1) It is absolutely necessary to draw one or more cartoons to make it easier for the audience to read and follow the paper.
Response: We agree with the reviewer and have drawn two cartoons that have been included as Figures 1 and 2 in the revised version. The first one introduces the terms autocrine, paracrine and intracrine TGF-β signaling, while the second one summarizes the role of autocrine TGF-β in therapy resistance (section 3.9.).
2) As stated in the introduction, tgfb has a great impact on immune cells and CAF. It would be great and more explanatory if the author could dedicate a section to this as well.
Response: This is a good suggestion. However, we have recently published a comprehensive review on this topic, which has been cited (Ref. 7). In the present manuscript we have already discussed a couple of studies with CAFs. We feel that extending this to an entire section would be beyond the scope of this section, mainly because the interactions between cancer cells and CAFs are of paracrine nature. However, the focus of our manuscript should remain on autocrine interactions.
Reviewer 2 Report
TGF-β is an important factor involved in a variety of processes, including immune responses / inflammation and cancer development. The complex metabolism of TGF-β ligands and complicated signal transduction make TGF-β research often confusing but intriguing. The review is devoted to the autocrine and paracrine effects of TGF-β in cancer. In addition, the review explores the limitations of the experimental analysis of TGF-β effects. The involvement of TGF-β in various processes of carcinogenesis and its interaction with factors involved in the development of cancer are also discussed.
The review is well written and covers the main ideas and recent research on TGF-β and cancer. However, I would like to make a few small comments that the author needs to pay attention to.
Major comments.
- Line 76-79 please rephrase the sentence to make it clearer.
- Line 141: The transition between paragraphs is not clear. The authors begin to talk about the Neo mutation (Oncogene Neo) do not introduce the reader to what Oncogene Neo is and why the authors associate it with TGFb pathways. it becomes clearer when you read further, but still, I would like to clarify why Neo is so important.
- Lines 157-162: Please rephrase this paragraph to make it clearer for the reader.
- Lines 203-208: It is not clear why the authors inserted this paragraph, what does it illustrate in the context of tumor development? If the authors want to report on the effect of TGFb on endothelial development, it's better to say so directly. This paragraph dropped out of the course of the text.
- Line233-235: The author introduces pathway target (E3 ubiquitin-protein-ligase XIAP), but does not explain why this target in this paragraph is interesting for the reader.
- Despite the fact that, in general, the information presented in the text is very interesting, but the text is sometimes difficult to read due to the systemic complexity of the sentences and the lack of a visual explanation (diagram, drawing, figures, etc).
Minor revision
- Line 13: promote -> promotes
- When a term or abbreviation is used for the first time, it has to be explained, please (e.g. Line 155: “PyMT”; line 360: “SHON” )
- Lines 166-168: please clarify the sentence.
- Line 238: What did the author mean by saying “author’s own laboratory”?
- Line 690: “… which resulted in a MET” it is unclear what event leads to MET in this sentence.
- Line 760-761: “…. some even TβRII for inhibition” Please clarify the sentence. Which is especially difficult when selecting TβRII inhibitors?
Author Response
Dear Editor, dear Arielia,
Please find enclosed the revised version of our manuscript (Manuscript ID: ijms-1073424). We thank the reviewers for their enthusiastic comments and have done our best to incorporate their suggestions in the revised version (highlighted in “track changes” mode). We believe that this has considerably improved its quality. We are looking forward to its final acceptance in the IJMS.
With warm regards,
Hendrik Ungefroren
TGF-β is an important factor involved in a variety of processes, including immune responses / inflammation and cancer development. The complex metabolism of TGF-β ligands and complicated signal transduction make TGF-β research often confusing but intriguing. The review is devoted to the autocrine and paracrine effects of TGF-β in cancer. In addition, the review explores the limitations of the experimental analysis of TGF-β effects. The involvement of TGF-β in various processes of carcinogenesis and its interaction with factors involved in the development of cancer are also discussed.
The review is well written and covers the main ideas and recent research on TGF-β and cancer. However, I would like to make a few small comments that the author needs to pay attention to.
Major comments.
- Line 76-79 please rephrase the sentence to make it clearer.
Response: This sentence has been rephrased and shortened to enhance clarity. A broader discussion of this issue is presented in the first paragraph of section 2.1.
- Line 141: The transition between paragraphs is not clear. The authors begin to talk about the Neo mutation (Oncogene Neo) do not introduce the reader to what Oncogene Neo is and why the authors associate it with TGFb pathways. it becomes clearer when you read further, but still, I would like to clarify why Neo is so important.
Response: As requested, we have started the paragraph with an introduction of Neu/erbb2. It is well established that EGFR signaling crosstalks with TGF-β signaling.
- Lines 157-162: Please rephrase this paragraph to make it clearer for the reader.
Response: As requested, this paragraph has been rephrased.
- Lines 203-208: It is not clear why the authors inserted this paragraph, what does it illustrate in the context of tumor development? If the authors want to report on the effect of TGFb on endothelial development, it's better to say so directly. This paragraph dropped out of the course of the text.
Response: We agree with the reviewer and have removed this paragraph including Ref. 33 from the manuscript.
- Line 233-235: The author introduces pathway target (E3 ubiquitin-protein-ligase XIAP), but does not explain why this target in this paragraph is interesting for the reader.
Response: An explanation has been added (XIAP decreases protein levels of the tumor suppressor PTEN).
- Despite the fact that, in general, the information presented in the text is very interesting, but the text is sometimes difficult to read due to the systemic complexity of the sentences and the lack of a visual explanation (diagram, drawing, figures, etc).
Response: We totally agree with the reviewer. Since the inclusion of one or two figures was also requested by reviewer 1, we have added two figures; one that introduces the terms autocrine, paracrine and intracrine TGF-β and another one in section 3.9. that summarizes the role of autocrine TGF-β in therapy resistance.
Minor revision
- Line 13: promote -> promotes
Response: Done
- When a term or abbreviation is used for the first time, it has to be explained, please (e.g. Line 155: “PyMT”; line 360: “SHON” )
Response: „PyMT“ has already been defined in the second paragraph of section 2.1. (line 114 in the original version). All terms have now been defined at first mention.
- Lines 166-168: please clarify the sentence.
Response: This sentence has been rephrased and simplified.
- Line 238: What did the author mean by saying “author’s own laboratory”?
Response: I refer here to data from my own group. The term „laboratory“ has been replaced by „group“.
- Line 690: “… which resulted in a MET” it is unclear what event leads to MET in this sentence.
Response: This has been rephrased to „…, which resulted in reversal of EMT“.
- Line 760-761: “…. some even TβRII for inhibition” Please clarify the sentence. Which is especially difficult when selecting TβRII inhibitors?
Response: This part of the sentence has been removed for the sake of clarity.
Additional changes made:
- Three new references have been added (Refs. 10,11, and 24 in the revised version).
- All the sentences with high similarity to published papers have been rephrased and highlighted in the track changes mode (in this case the yellow color was removed).
- The last paragraph of section 3.9. (lines 690-696 in the original version) has been moved to section 3.8.